# Molecular Identification and Immunity Functional Characterization of *Lmserpin1* in *Locusta migratoria manilensis*

**DOI:** 10.3390/insects12020178

**Published:** 2021-02-18

**Authors:** Beibei Li, Hongmei Li, Ye Tian, Nazir Ahmed Abro, Xiangqun Nong, Zehua Zhang, Guangjun Wang

**Affiliations:** 1State Key Laboratory of Plant Diseases and Insect Pests, Institute of Plant Protection, Chinese Academy of Agricultural Science, Beijing 100081, China; 82101182322@caas.cn (B.L.); 1917737420@139.com (Y.T.); nongxiangqun@caas.cn (X.N.); zhangzehua@caas.cn (Z.Z.); 2Scientific Observation and Experimental Station of Pests in Xilingol Rangeland, Ministry of Agriculture and Rural Affairs, Xilinhot 026000, China; 3MARA-CABI Joint Laboratory for Bio-safety, Institute of Plant Protection, Chinese Academy of Agricultural Science, Beijing 100193, China; h.li@cabi.org; 4College of Plant Protection, Hebei Agricultural University, Baoding 071001, China; 5Department of Entomology, Sindh Agriculture University, Tando Jam 70060, Pakistan; 2017y90100019@caas.cn

**Keywords:** locust, *metarhizium anisopliae*, *lmserpin1*, immune related gene

## Abstract

**Simple Summary:**

Insect serpins play a vital role in the defense mechanism of insects, especially in the Toll pathway and PPO (prophenoloxidase) cascade. In this study, we provided an answer to the open question of whether serpin1 was involved in the humoral immune responses of *Locusta migratoria manilensis*. We identified a new *Lmserpin1* gene from *L. migratoria manilensis* and investigated its expression profiles in all examined stages and tissues. Meanwhile, by interfering with *Lmserpin1* gene, we examined the mortality of *L. migratoria manilensis* under *Metarhizium anisopliae* infection, as well as the activities of protective enzymes and detoxifying enzymes and the expression level of three immune-related genes (*PPAE* (prophenoloxidase-activating enzyme), *PPO*, and *defensin*). The results indicated that *Lmserpin1* gene up-regulated the immune responses of *L. migratoria manilensis* and inhibited the infection of *M. anisopliae.* Our results are of great importance for better understanding of the mechanism characterization of *Lmserpin1* in *L. migratoria manilensis*.

**Abstract:**

Serine protease inhibitors (Serpins) are a broadly distributed superfamily of proteins that exist in organisms with the role of immune responses. *Lmserpin1* gene was cloned firstly from *Locusta migratoria manilensis* and then was detected in all tested stages from eggs to adults and six different tissues through qRT-PCR analysis. The expression was significantly higher in the 3rd instars and within integument. After RNAi treatment, the expression of *Lmserpin1* was significantly down-regulated at four different time points. Moreover, it dropped significantly in the fat body and hemolymph at 24 h after treatment. The bioassay results indicated that the mortality of *L. migratoria manilensis* treated with dsSerpin1 + Metarhizium was significantly higher than the other three treatments. Furthermore, the immune-related genes (*PPAE*, *PPO*, and *defensin)* treated by dsSerpin1 + Metarhizium were significantly down-regulated compared with the Metarhizium treatment, but the activities of phenoloxidase (PO), peroxidase (POD), superoxide dismutase (SOD), glutathione S-transferase (GST), and multifunctional oxidase (MFO) were fluctuating. Our results suggest that *Lmserpin1* plays a crucial role in the innate immunity of *L. migratoria manilensis. Lmserpin1* probably took part in regulation of melanization and promoted the synthesis of antimicrobial peptides (AMPs).

## 1. Introduction

Host and pathogens are an interactive relationship locked in a perpetual evolutionary marathon due to the evolving microorganisms and adaptable host immune system [1,2]. Innate immunity is a fundamental self-protection system that evolved in all animals to protect against exogenous pathogenic signal. Unlike mammals, insects rely solely on innate immune mediators to prevent infection [3,4]. Pathogen-associated molecular patterns (PAMPs) activate host cell-surface receptors (pathogen-recognition receptors or PRRs) when infection occurs. These signal cascades activate both cellular and humoral innate immune mechanisms [5,6,7].

Innate humoral immune responses are initiated through Toll and IMD(immune deficiency) signaling pathways during the early invasion period leading to the synthesis and secretion of antimicrobial peptides (AMPs) [8]. Once insects are invaded by external microorganisms, prophenoloxidase-activating enzyme (PPAE) activate the prophenoloxidase cascade reaction (PPO cascade), and specific serine protease activate the phenoloxidase (PO) which, together, with tyrosine hydroxylase then catalyze blood coagulation and melanin formation, participating in the immune defense response of the body [9]. Extracellular parts of both the Toll signaling pathway and PPO activation system employ cascades of serine proteinases to amplify the initial invading signal, ultimately resulting in rapid and efficient responses to threats. However, to avoid excessive inflammation and immunopathology, this signaling cascade must be well-regulated. Serine protease inhibitors (serpins), as serine or cysteine protease inhibitors, can regulate serine protease cascades to maintain homeostasis in the organisms [10].

Serpins are the largest and most widely distributed superfamily of protease inhibitors, presenting in animals, plants, and microorganisms [11,12,13]. To date, at least 23 different families of serpins have been described and at least 12 studied in insects, such as *Drosophila melanogaster* [14], *Maduca sexta* [15], *Bombyx mori* [16], and *Choristoneura fumiferna* [17]. Serpins are 40–50 kDa plasma proteins with approximately 400 amino acid residues in length. The majority of arthropod serpins regulate many innate immune responses, including the hemolymph serine protease cascade [18,19,20,21]. Their main function is to prevent excessive inflammation and immunopathology through irreversibly regulating the activity of serine proteases or inactivating proteases by forming covalent complexes [22]. For example in *M. sexta*, *Msserpin4*, and *Msserpin5*, PPO activation is inhibited by inhibiting the upstream hemolymph phase, and *Ms*serpin6 inhibits PPO activation via the regulation of prophenoloxidase-activation protease 3 (PAP3) and hemolymph protease 8 (HP8) [23,24]. In *D. melanogaster*, *Dmserpin43Ac* regulates Toll and PPO pathways in response to fungal and bacterial infections [25]. It has been described that 13 serpins (1–7, 11–13, 21, 28, and 32) in *B. mori*, as protease inhibitors, can regulate extracellular serine proteases involved in innate immune responses such as PPO activation, spätzle processing, and embryonic development by Toll pathway activation [26,27,28].

The migratory locust, *L. migratoria manilensis* is an economically important insect pest that causes serious crop losses and pasture damage in China, South-East Asia, and the Pacific region [29]. A previous study found that the *M. anisopliae* strains had a strong virulence to *L. migratoria manilensis* [30]. To better understand the immune mechanisms of *L. migratoria manilensis* and examine their response to a variety of pathogen challenges, transcriptome sequencing and analysis were performed on migratory locusts. A total amino acid sequences of seven serpin genes (*serpin1* to *serpin7*) were obtained [31]. Phylogenetic tree analysis showed that *Lmserpin1* was similar to *serpinB9*, which was involved in the immune response [32]. It is speculated that *Lmserpin1* may be involved in the immunity of the migratory locust through structural prediction. The expression pattern of *Lmserpin1* and its function in *L. migratoria manilensis* were analyzed. This study will promote further exploration of the immune mechanism of the *Lmserpin1* gene in *L. migratoria manilensis*.

## 2. Materials and Methods

### 2.1. Insect Rearing

Eggs of *L. migratoria manilensis* were collected from Cangzhou in Hebei Province, China. Eggs mixed with soil were incubated under 27 ± 0.5 °C, 60 ± 5% RH in growth cabinets (PRX-350B-30) until hatching. Freshly emerged locust hoppers were transferred into wooden boxes (60 cm × 50 cm × 60 cm, 27 ± 0.5 °C, 14:10 L:D photoperiod), and fed on wheat seedlings.

### 2.2. Expression Analysis of Lmserpin1 Gene in Different Tissues and Developmental Stages

Total RNA was extracted from different tissues (midgut, testis, integument, metapedes (the leaping feet), fat body, and hemolymph) and developmental stages (egg, 1st, 2nd, 3rd, 4th, 5th, and adult) using the Trizol reagent (Invitrogen Corp., Carlsbad, CA, USA) according to the manufacturer’s instructions. The concentration and integrity of RNA were measured at an absorbance ratio of A260/280 and A260/230 using a NanoDrop 2000 spectrophotometer (Thermo Scientific TM, Waltham, MA, USA), and by 1.0% agarose gel electrophoresis, respectively. The first-strand cDNA was synthesized using PrimeScript™ One Step RT-PCR Kit Ver. 2 (Takara, Dalian, China) according to the manufacturer’s instructions. cDNA preparations from different samples were used as templates for qRT-PCR analysis of the *Lmserpin1* gene expression level using a specific primer pair, qPCR-serpin1-F and qPCR-serpin1-R (Table 1). qRT-PCR reactions were performed with the SYBR Premix ExTaq™ (TaKaRa, Dalian, China) in 20 μL of reaction volume with 1μL of cDNA, 10 μL of SsoFast SYBR-Green Mix, 1 μL of each primer (20 μM), and 7 μL of double distilled water (ddH_2_O), on the ABI 7500 Real-Time PCR System (Applied Biosystems, Foster City, CA, USA). qRT-PCR conditions were as follows: Denaturation at 95 °C 60 s, followed by 40 cycles of amplification (95 °C 15 s, 60 °C 50 s). Actin was used as an external reference control. Each plate was repeated three times in independent runs for all reference and selected genes. Gene expression was evaluated by the 2^−ΔΔCT^ method [33]. The relative expression of developmental stages was calculated and normalized to the expression at the egg stage, of which different tissues were calculated and normalized to the expression at the hemolymph [34].

### 2.3. Cloning of the Lmserpin1 Gene and RNA Interference

Using cDNA of 3rd instars as the templates and serpin1-R and serpin1-F as gene-specific primers (Table 1), the open reading frame (ORF) of *Lmserpin1* was amplified by PCR under the following protocol: 95 °C for 5 min; 35 cycles of 95 °C for 30 s; 56 °C for 30 s; 72 °C for 90 s; and 72 °C for 10 min. The PCR products were recovered on 1% agarose gel and purified using a DNA purification Kit (Tiangen Biotech, Beijing, China). Purified PCR products were cloned into the pMD19-T vector and sequenced by the Shanghai Sangon Biological Co. LTD to verify the cloned fragments.

RiboMAXTM System-T7 (Promega, Madison, WI, USA) was used to generate double-stranded *serpin1* (dsSerpin1) RNA by in vitro transcription following the manufacturer’s instructions with dsSerpin1-F and dsSerpin1-R (Table 1) as primers and plasmid DNA containing the *Lmserpin1* gene as the template. The PCR reaction conditions were as follows: 94 °C for 10 min; 35 cycles of 94 °C for 30 s; 58 °C for 30 s; 72 °C for 90 s; and 72 °C for 10 min. dsSerpin1 RNA was then diluted to appropriate concentration using ddH_2_O and stored at −80 °C for later use.

3rd instars were starved for 12 h in advance, dsSerpin1 RNA (5 μg) was then injected into the ventral area between the second and third abdominal segments [35], and ddH_2_O was used as the control. There were three replicates and 20 individuals for each replicate. *Lmserpin1* transcript levels in the whole body were tested at 6 h, 24 h, 48 h, and 72 h after injection and in different tissues (fat body, midgut, testis, and hemolymph) at 24 h through qRT-PCR with the specific primer pair, qPCR-serpin1-F, and qPCR-serpin1-R (Table 1).

### 2.4. Expression Analysis of Lmserpin1 after Metarhizium Anisopliae Infection

To determine the expression of *Lmserpin1* after *Metarhizium anisopliae* infection, the total RNA was extracted and cDNA was prepared as described above at 6, 12, 24, 48, and 72 h after infection. The mRNA level of *Lmserpin1* was analyzed by qRT-PCR. The transcriptional level of *Lmserpin1* was normalized to *L. migratoria manilensis* actin. The experiment was conducted in triplicate.

### 2.5. Bioassay

Conidia of *M. anisopliae* was from the Institute of Plant Protection, Chinese Academy of Agricultural Sciences. The concentration of conidia was adjusted to 2.5 × 10^8^ spores/g using control bait which was prepared with sterile wheat bran containing 5% vegetable oil, and then used as the treatment bait [36]. dsSerpin1 RNA (l μg/μL) was prepared as for as step 2.3. The four different treatments were (1) inject 5 μL of ddH_2_O with the treatment bait, (2) inject 5 μL of dsSerpin1 RNA with the control baits, (3) inject 5 μL of dsSerpin1RNA with the treatment baits, and (4) inject 5μL of ddH_2_O with the control baits (Table 2). All of the 3rd instars were starved for approximately 12 h before treatments. For each treatment, there were five replicates and 30 individuals for each replicate. All baits were replaced by fresh wheat seedlings after 24 h, and dead instars were recorded daily for 12 days.

### 2.6. Enzyme Activity Assay

3rd instars *L. migratoria manilensis* were treated using the method described above, and samples were taken for 5 days continuously after treatments. The samples of the *L. migratoria manilensis* body were homogenized with 2 mL of 0.1 moL/L phosphate buffer solution (PBS), and then centrifuged at 11,000 rpm for 10 min at 4 °C. Resulting supernatants were transferred to a clean eppendorf tube and centrifuged again at 11,000 rpm for 15 min at 4 °C. The final supernatants were used for proteins and enzyme detection.

The activities of phenoloxidase (PO), multifunctional oxidase (MFO), and glutathione transferase (GSTs) were measured according to previously described protocol using Softmax Pro 6.1 software [37,38,39]. Superoxide dismutase (SOD) and peroxidase (POD) activities were measured by the manufacturer’s instruction (Nanjing Jiancheng Biochemical Institute, Nanjing, China). Moreover, protein content was measured using the Bradford method [40], with bovine serum albumin (BSA) as the standard.

### 2.7. Quantitative Analysis of Immune Related Gene Expression

To determine the expression of the immune-related gene (*PPAE*, *PPO*, and *defensin*), fat body was collected from 3rd instars of *L. migratoria manilensis* at 24 h after the treatments. Total RNA was extracted and cDNA was prepared as described above. The mRNA level of *PPAE*, *PPO*, and *defensin* was analyzed by qRT-PCR using a specific primer pair (Table 1). Each sample was analyzed by the threshold cycle (CT). The data were shown as means ± standard deviations. Actin was used as an external reference control. The experiment was conducted in triplicate.

### 2.8. Statistical Analysis

Statistical analysis was done using the independent samples t-test and one-way ANOVA followed by post hoc test (Duncan’s new multiple range test). *p* < 0.05 was considered statistically significant, and the abc and asterisks on the bars in the figures represent significant differences among the treatments and control. All the studied traits and data were analyzed using SPSS, version 19.0 (SPSS Inc., Chicago, IL, USA) and PRISM, version 6.01 (GraphPad Software Inc., San Diego, CA, USA).

## 3. Result

### 3.1. Expression of Lmserpin1 in Different Tissues and Developmental Stages of L. migratoria manilensis

A full-length cDNA representing a putative *L. migratoria manilensis* serpin was identified and named *Lmserpin1*. The full-length cDNA of *Lmserpin1* consists of 1228 bp containing 1040 bp open reading frame (ORF) and translates into a 347 amino acids protein with a single serpin domain (spanning from residues 21–343). The calculated molecular mass of the *Lmserpin1* protein is 35.6 kDa with an estimated pI of 6.19.

The tissue distribution and developmental stages expression of *Lmserpin1* in *L. migratoria manilensis* were analyzed by qRT-PCR (Figure 1). The results indicated that *Lmserpin1* was consistently expressed in all life stages (Figure 1A), and its expression level increased with the development of *L. migratoria manilensis*, reaching peak at 3rd instars. The *Lmserpin1* expression levels at the 1st, 2nd, and 3rd instar were 1.224, 1.607, and 1.841 times of that in egg respectively. Moreover, *Lmserpin1* expression at 3rd instars was significantly higher than the other stages (*p* < 0.05). While *Lmserpin1* expression at 4th instars and adults was significantly lower than other stages (*p* < 0.05), and their expression levels was only 0.175 and 0.597 times that in the egg, respectively. *Lmserpin1* expression at 5th instars was 1.122 times that in the egg respectively, which was no significantly different with egg. Therefore, we selected 3rd instars as the target stage to show the response to pathogenic bacteria.

Furthermore, the tissues distribution of *Lmserpin1* was also observed in midgut, testis, integument, metapedes, fat body, and hemolymph, respectively (Figure 1B). The results indicated that the transcriptional level of *Lmserpin1* in integument was the highest, with 28.825 times that of the hemolymph, meanwhile in metapedes and fat body it was 11.996 and 21.256 times that of the hemolymph (*p* < 0.05), respectively. While the expression level of *Lmserpin1* in testis and midgut had no difference with the hemolymph (*p* > 0.05).

### 3.2. The Expression of Lmserpin1 of L. migratoria manilensis in 3rd Instars after RNAi

Analysis of variance revealed significant differences for the relative expression of *Lmserpin1* in *L. migratoria manilensis.* The expression of *Lmserpin1* was significantly lower in treated samples compared to that of the control (inject ddH_2_O) at 6 h, 24 h, 48 h, and 72 h after treatment, and it was only 0.160 times of that of the control at 24 h (Figure 2A, *p* < 0.05). Therefore, we selected 24 h after treatment as the point in time to show the response of different tissue to *Lmserpin1* RNAi.

Moreover, the expression of *Lmserpin1* in different tissue, after it was silenced through RNAi, was significantly down-regulated at 24 h after treatment, in particular in fat body and hemolymph, it was only 0.0101 and 0.0171 times that of the control (*p* < 0.05). The result showed that the *Lmserpin1* gene was interfered efficiently in fat body. Therefore, we selected fat body as the target tissues to show the response to pathogenic bacteria.

### 3.3. The Expression of Lmserpin1 with Application of M. anisopliae

After infection with *M. anisopliae* spores, the expression of *Lmserpin1* of *L. migratoria manilensis* was significantly higher than the control group in the early stage (Figure 3). It was 6.285, 5.483, and 1.558 times that of the control at 6 h, 12 h, and 24 h after treatment, respectively. While there was no significant difference between the control and treatment at 48 h and 72 h.

### 3.4. Mortality Comparison under Different Bioassay Treatments

With the application of *M. anisopliae*, the mortality of *L. migratoria manilensis* increased to 67.78% at the 12th day after infection, and from the 5th to 12th day, all of them were significantly higher than that of the control (*p* < 0.05) (Figure 4). Injection of dsSerpin1 RNA killed only 18.32% of the *L. migratoria manilensis*, with no difference with the control which was only 8.77% at the 12th day. The maximum mortality (94.31%) was recorded when dsSerpin1RNA and Metarhizium was applied in combination, and it was significantly higher than that of Metarhizium used alone and the control from day 5 (*p* < 0.05). Moreover, the mortality was consistently increased in all the treatments with extending the time, from 5 to 12 days of treatment, the mortality of Metarhizium group was substantially significantly higher than the control group, while significantly lower than the dsSerpin1 + Metarhizium group (*p* < 0.05).

### 3.5. The Enzyme Activity of L. migratoria manilensis among Different Treatment Groups

#### 3.5.1. Effects of Different Treatment Groups on PO Activity of *L. migratoria manilensis*

The activity of PO treated by Metarhizium showed a trend of increasing then decreasing with days, while that treated by the dsSerpin1 + Metarhizium showed a trend of decreasing then increasing with days (Figure 5). Treated by Metarhizium, the activity of PO was significantly lower than that of the control, with the exception of the 3rd day of sampling (*p* < 0.05), and was not significantly different from the control on the 3rd day of sampling. Treated by dsSerpin1 + Metarhizium, the activity of PO was significantly higher than that of the control on the 4th day while lower than that of the control on the other days of sampling (*p* < 0.05). Moreover, it significantly higher than the Metarhizium treatment group on the 4th day, and was significantly lower than the Metarhizium treatment group on the 2nd and 3rd days (*p* < 0.05), which is not significantly different from the Metarhizium treatment group on other days.

#### 3.5.2. Effects of Different Treatment Groups on POD Activity of *L. migratoria manilensis*

The activity of POD treated by Metarhizium or by the dsSerpin1 + Metarhizium showed a trend of decreasing then increasing and then decreasing with the days (Figure 6). Treated by Metarhizium, the activity of POD was significantly higher than that of the control on the 1st day, but significantly lower than that of the control on the 2nd and 5th day of sampling (*p* < 0.05), and it was not significantly different from the control on other days. Treated by dsSerpin1 + Metarhizium, the activity of POD was significantly higher than that of the control on the 1st day while lower than that of the control on the 2nd, 4th, and 5th days (*p* < 0.05). Moreover, it was significantly higher than the Metarhizium treatment group on the 1st and 5th days, and was significantly lower than the Metarhizium treatment group on the 4th day (*p* < 0.05), which not significantly different from the Metarhizium on other days.

#### 3.5.3. Effects of Different Treatment Groups on SOD Activity of *L. migratoria manilensis*

The activity of SOD treated by Metarhizium or by the dsSerpin1 + Metarhizium decreased with days (Figure 7). Treated by Metarhizium, the activity of SOD was significantly higher than that of the control on the 1st day, but significantly lower than that of the control on the 5th day (*p* < 0.05), and it was not significantly different from the control on other days. Treated by dsSerpin1 + Metarhizium, the activity of SOD on the first two days were significantly higher than that of the control and the Metarhizium, but was significantly lower than that of the two group on the 4th day (*p* < 0.05). There was no significantly difference found among them on the 3rd and 5th day (*p* > 0.05).

#### 3.5.4. Effects of Different Treatment Groups on MFO Activity of *L. migratoria manilensis*

The MFO activity of the group treated with the Metarhizium showed a trend of increasing then decreasing with days, while the activity of MFO in the dsSerpin1 + Metarhizium treatment group showed a trend of decreasing then increasing with days (Figure 8). Treated by Metarhizium, the activity of MFO was significantly higher than that of the control on the 2nd and 3rd days of sampling, but significantly lower than that of the control on the 4th and 5th days of sampling (*p* < 0.05), and was not significantly different from the control he 1st day (*p* > 0.05). Treated by dsSerpin1 + Metarhizium, the activity of MFO was significantly lower than that of the control on the 4th and 5th days, and was not significantly different from the control on other days (*p* < 0.05). Moreover, it was significantly lower than the Metarhizium treatment group on the on the 2nd, 3rd, and 4th days of sampling, and was not significantly different from the Metarhizium treatment group on other days (*p* > 0.05).

#### 3.5.5. Effects of Different Treatment Groups on GSTs Activity of *L. migratoria manilensis*

The activity of GSTs in the Metarhizium treatment group showed a trend of decreasing then increasing with days, while that in the dsSerpin1 + Metarhizium treatment group showed a trend of decreasing then increasing, then decreasing with days (Figure 9). Treated by Metarhizium, the activity of GSTs was significantly lower than that of the control on the 3rd and 5th days of sampling (*p* < 0.05), and was not significantly different from the control on other days (*p* > 0.05). Treated by dsSerpin1 + Metarhizium, the activity of GSTs was significantly higher than that of the control on the 1st and 4th days while lower than that of the control on the 3rd and 5th day (*p* <0.05), and was not significantly different from the control on the 2nd day. Moreover, the activity of GSTs in the dsSerpin1 + Metarhizium treatment group was significantly higher than the Metarhizium treatment group on the 1st and 4th days, and was significantly lower than the Metarhizium treatment group on the 5th day (*p* < 0.05), and was not significantly different from Metarhizium treatment group on other days.

### 3.6. Effect of RNAi on mRNA Levels of Immune Related Pathway Gene

Analysis of variance revealed that the relative expression levels of *PPAE*, *PPO*, and *defensin* mRNA in *L. migratoria manilensis* treated by dsSerpin1 + Metarhizium showed a decreased trend compared to that treated by Metarhizium alone (Figure 10). The expression level of *PPAE* treated by the Metarhizium was significantly higher than that of other three treatments, while that treated by dsSerpin1 + Metarhizium was significantly lower than the dsSerpin1 treatment and there was no significantly difference found comparing with the control (Figure 10A, *p* < 0.05).

The expression level of *PPO* treated by the Metarhizium was significantly higher than that of other three groups, while treated by dsSerpin1 + Metarhizium, *L. migratoria manilensis* sharply suppressed the expression of *PPO,* so that there were no significant differences observed between the dsSerpin1 + Metarhizium treatment and the dsSerpin1 treatmeat as well as the control treatment respectively (Figure 10B, *p* < 0.05).

For the expression level of *defensin*, when treated by the Metarhizium group, it was significantly higher than that of the control, while treated by dsSerpin1 + Metarhizium, it was significantly lower than that of the independent treatment of Metarhizium, while no such significant differences were observed with the control treatment (Figure 10C, *p* < 0.05).

## 4. Discussion

When infected by *M. anisopliae*, we found that *Lmserpin1* was expressed with a higher level at 6 h, 12 h, and 24 h after treatment, which means that *Lmserpin1* gene was involved in the immune response of migratory locusts (Figure 3). To gain a better understanding of the expression profile of *Lmserpin1* in different instars of *L. migratoria manilensis*, we cloned the *Lmserpin1* gene from *L. migratoria manilensis* and investigated its transcription by qRT-PCR. The results showed that the *Lmserpin1* gene was expressed in all stages with the highest expression level in 3rd instars (Figure 1A). Serpins had been found to be associated with the innate immune system [11], thus we inferred that the nymph of *L. migratoria manilensis* in the 3rd instars had the highest resistance to pathogens, and we selected the 3rd instars as the optimal stage to show the response to pathogenic bacteria. Furthermore, we found that *Lmserpin1* was expressed with a higher level in integument and fat body (Figure 1B). The higher expression level of *Lmserpin1* in the integument might be due to direct contact between the tissue and *M. anisopliae*, which was helpful for successfully preventing the infection of fungi. Meanwhile, a higher expression level in the fat body may be due to the main tissue of innate immune in *L. migratoria manilensis.* When the *Lmserpin1* gene was interfered with, the expression level was the lowest at 24 h after treatment, with a relative level no more than 20% compared to the control, which indicated that *Lmserpin1* was successfully interfered (Figure 2A). In addition, the interference efficiency was the highest in fat body and hemolymph (Figure 2B). Previous studies have shown that the fat body and hemolymph were mediators of cellular immunity in arthropods and contributed to the humoral branch by expressing a large number of secreted immune factors [41]. Thus we chose the fat body as the optimal tissue to detect the effect of *Lmserpin1* interferrence in the immune response. *M. anisopliae* is the most important entomopathogenic fungus with potential use against many insect pests, including locusts and grasshoppers [42]. Previous studies have shown that the Metarhizium adhesion-like protein 1(Mad1) with Metarhizium can cause a higher mortality in migratory locusts than Metarhizium alone [43]. In our study, the mortality of *L. migratoria manilensis* treated with dsSerpin1 + Metarhizium was 94.31%, while being treated with Metarhizium alone was 67.78% (Figure 4). This result indicated that interfering with *Lmserpin1* could effectively enhance the infection of Metarhizium and increase the mortality of *L. migratoria manilensis*, however, it means that *Lmserpin1* can inhibit the infection of *M. anisopliae*. For all we know, extracellular proteinase, which is a kind of main virulence factor, plays a vital role during the infection of *M. anisopliae*. We inferred that *Lmserpin1*, as a serine protease inhibitor, may have inhibited the activity of extracellular proteinase from *M. anisopliae*, decreased its virulence, ultimately reducing the insecticidal efficiency of locusts caused by *M. anisopliae* infection.

When pathogens invade, humoral immunity will stimulate the synthesis of melanin to remove foreign pathogens [44]. PO is an important factor in catalyzing melanization [45]. Serpins acting as regulators of melanization response have been reported in various insects [21]. Such as in *M. sexta*, *Mxserpin3*, *Mxserpin4*, *Mxserpin5*, and *Mxserpin6* has been found to be involved in the regulation of PPO cascade [23,24,46]. This study showed that *PPAE* and *PPO* genes were significantly enhanced following *M. anisopliae* infection (Figure 10A,B), while interference with the *Lmserpin1* gene significantly reduced this two gene expression in the immune-related pathway when a locust is infected. Additionally, the activity of PO in the dsSerpin1 + Metarhizium combined treatment group was significantly reduced than the Metarhizium group in the early stage (Figure 5). Those results showed that interference of the *Lmserpin1* gene could effectively inhibit the expression of melanization-related pathway genes and the activity of PO, ultimately decreasing the ability of locust to resist pathogenic bacteria infection, which means *Lmserpin1* could enhance the melanization reaction of *L. migratoria manilensis.* It has been described that the *Dmserpin27A* of *D. melanogaster* can inhibit the activities of *PPAE* and PO at the site of injury or infection to prevent the insect from excessive melanization [14], and *Dmserpin8* can inhibit the PO activity in the hemolymph of *Penaeus Monodon* and participates in its own melanization [47]. The function of *Lmserpin1* was different from that of *Dmserpin27A* and *Dmserpin8* to a certain extent, further research is needed.

The invasion of *M. anisopliae* leads to changes of different enzyme activities in migratory locusts, which destroys the balance of the enzyme system. When the destruction of homeostasis reaches a certain degree, the migratory locust begins to die [48]. SOD and POD can scavenge superoxide anion free radicals (O^2−^) and hydrogen peroxide (H_2_O_2_), which plays an important role in insect resistance to pathogenic bacteria. It has been reported that the invasion of Metarhizium can induce the increasing of O^2−^ concentration from locusts, enhancing the activity of SOD and POD to clear massive O^2−^ and excess H_2_O_2_ [49,50]. In our experiment, the activity of SOD and POD in the prior period was significantly increased when locusts were treated with *M. anisopliae*, and after interference of the *Lmserpin1* gene, the SOD enzyme activity was significantly higher than that without interference in the early stage of infection (Figure 6 and Figure 7), which indicated that the balance of the protective enzyme system of *L. migratoria manilensis* was damaged after *Lmserpin1* interference, it induced that the activity of the protective enzyme was over-increased, and ultimately mortality increased. It indicated that *Lmserpin1* could regulate the protective enzyme system of the insect body and regulate the resistance of *L. migratoria manilensis* to pathogenic bacteria. MFO and GSTs are two important families of enzymes in insects that participate in the detoxification of various xenobiotics and insecticides and play important roles in the resistance of insects to various insecticides [51]. Compared with Metarhizium, the dsSerpin1 + Metarhizium combined treatment could effectively reduce the activity of MFO and weaken the detoxification ability of *L. migratoria manilensis* (Figure 8). It indicated that the *Lmserpin1* gene could reduce the infection of *M. anisoplia*. By enhancing the detoxifying enzyme system and immunity response in migratory locusts. GSTs play a pivotal role in cellular antioxidant defenses against oxidative stress [52,53,54]. However, after the dsSerpin1 + Metarhizium combined treatment, the activity of GSTs is irregular, and further study is needed (Figure 9).

The effector AMPs produced from fat body constitute the humoral immune system in the fat body. It has been described that *Bmserpin15* protein in *B. mori* can reduce the transcript levels of the AMPs cecropin D, gloverin2, and moricin significantly [55]. The *Bmserpin28* in fat body of *B. mori* can inhibit significantly the transcription of AMPs [34]. While our qRT-PCR analysis result showed that after *L. migratoria manilensis* was infected by *M. anisopliae*, the expression of the *defensin* gene significantly decreased when the *Lmserpin1* gene was interfered with (Figure 10C). This result suggested that interference of the *Lmserpin1* gene could effectively inhibit the production of AMPs, which indicates that *Lmserpin1* could enhance the expression of AMPs genes to improve host humoral immune defensing.

In summary, after the *Lmserpin1* gene was interfered with, the activities of protective enzymes (SOD and POD), detoxification enzymes (GSTS and MFO), and PO in *L. migratoria manilensis* were fluctuating, and the expression levels of immune response related genes (PPAE, PPO, and defensin) decreased. These results show that *Lmserpin1* could increase immune responses in *L. migratoria manilensis* to resist the infection of *M. anisopliae*. However, there are many questions that remain to be answered. The mechanisms of *Lmserpin1*’s interaction with its target protease remain to be discovered. Future studies should focus on evaluating these signaling mechanisms to provide a clearer understanding of its possible functions in *L. migratoria manilensis*.

## 5. Conclusions

In this study, we monitored the activities of SOD, POD, PO, GSTs, and MFO, and evaluated the expression levels of immune-related genes of *L. migratoria manilensis* treated with dsSerpin1, Metarhizium, and dsSerpin1 + Metarhizium respectively to identify the immunity mechanism of *Lmserpin1*. Our results showed that the interference of *Lmserpin1* caused the activity of the protective enzyme over-increased and the detoxification enzyme suppressed and effectively reduced the expression of immune-related genes in *L. migratoria manilensis* which ultimately promoted the infection of Metarhizium, resulting in the *L. migratoria manilensis* to die acceleratively. These results indicated that *Lmserpin1* could increase the immune responses of *L. migratoria manilensis* to resist the infection of *M. anisopliae*.

## Figures and Tables

**Figure 1 insects-12-00178-f001:**
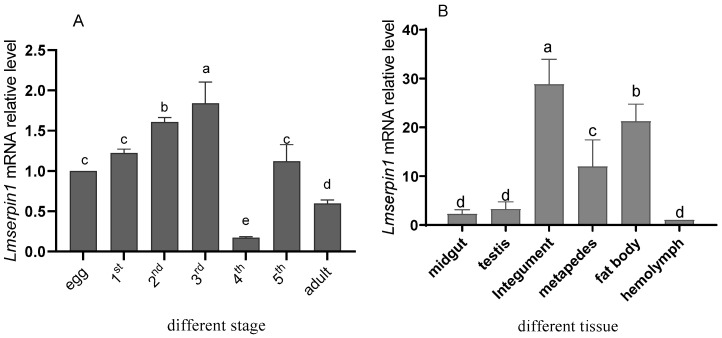
Expression of *Lmserpin1* in the different tissue and developmental stages of *L. migratoria manilensis*. (**A**) Relative mRNA level of the *Lmserpin1*gene in the development stages of *L. migratoria manilensis*, *Lmserpin1* mRNA level in the egg was attributed a value of 1. (**B**) Relative mRNA level of the *Lmserpin1* gene in the different tissue of *L. migratoria manilensis*, *Lmserpin1* mRNA level in the hemolymph was attributed a value of 1. Bars represent mean ± S.E. (n = 3). Bars labeled with different letters are significantly different (one-way ANOVA followed by Duncan’s test, *p* < 0.05).

**Figure 2 insects-12-00178-f002:**
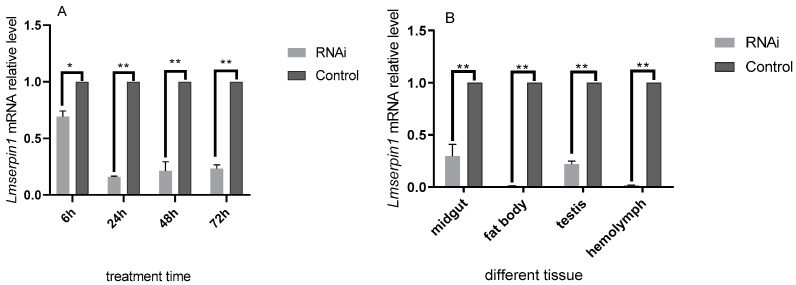
Relative quantitative expression of *Lmserpin1* mRNA in different time and different tissue of *L. migratoria manilensis after RNAi.* (**A**) Relative mRNA level of the *Lmserpin1*gene in the bodies of 3rd instars *L. migratoria manilensis* under treatment for 6 h, 24 h, 48 h, and 72 h. (**B**) Relative mRNA level of the *Lmserpin1* gene in the body, midgut, fat body, testis, and hemolymph after treatment by 24 h. Error probability of *p* < 0.05 by Student’s t-test, ** indicates a significant difference between the two groups; * indicates no significant difference between the two groups (Duncan’s method for multiple comparisons).

**Figure 3 insects-12-00178-f003:**
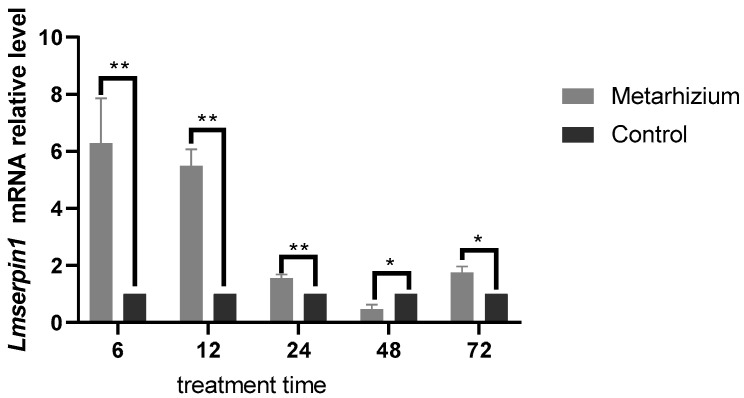
Relative quantitative expression of *Lmserpin1* mRNA of *L. migratoria* manilensis after being treated by *M. anisopliae*. Error probability of *p* < 0.05 by Student’s t-test, ** indicates a significant difference between the two groups; * indicates no significant difference between the two groups (Duncan’s method for multiple comparisons).

**Figure 4 insects-12-00178-f004:**
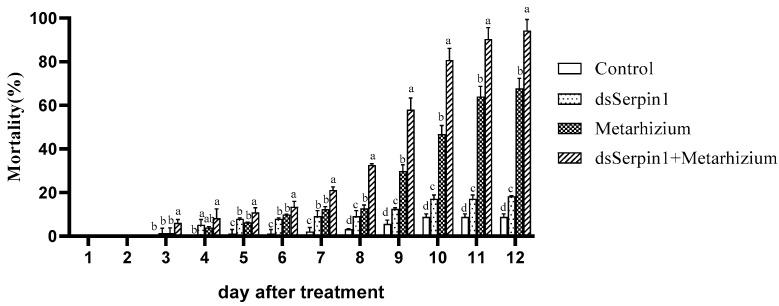
The mortality of *L. migratoria manilensis* days across 12 days after treatment, each treatments were 150 individuals. Bars represent mean ± S.E. (n = 3). Bars labeled with different letters are significantly different (one-way ANOVA followed by Duncan’s test, *p* < 0.05).

**Figure 5 insects-12-00178-f005:**
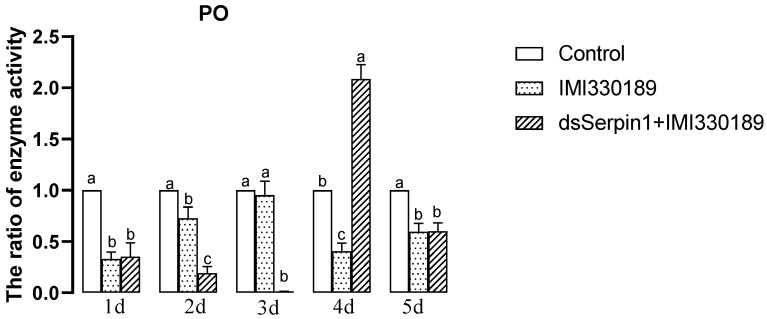
The phenoloxidase (PO) activity of *L. migratoria manilensis* on the 3rd instars after the application of different treatments. Bars represent mean ± S.E. (n = 3). The letters a, b, and c denote the differences between treatments, bars labeled with different letters are significantly different (one-way ANOVA followed by Duncan’s test, *p* < 0.05).

**Figure 6 insects-12-00178-f006:**
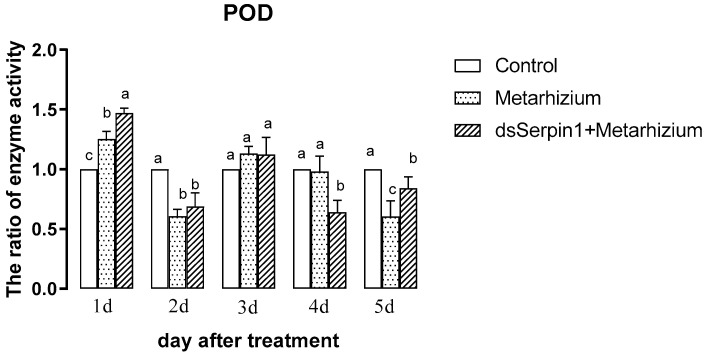
The peroxidase (POD) activity of *L. migratoria manilensis* from different treatments. Bars represent mean ± S.E. (n = 3). The letters a, b, and c denote the differences between treatments, bars labeled with different letters are significantly different (one-way ANOVA followed by Duncan’s test, *p* < 0.05).

**Figure 7 insects-12-00178-f007:**
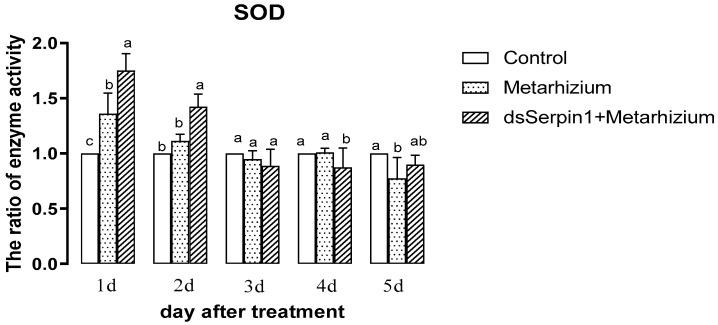
Superoxide dismutase (SOD) activity of *L. migratoria manilensis* from different treatments. Bars represent mean ± S.E. (n = 3). The letters a, b, and c denote the differences between treatments, bars labeled with different letters are significantly different (one-way ANOVA followed by Duncan’s test, *p* < 0.05).

**Figure 8 insects-12-00178-f008:**
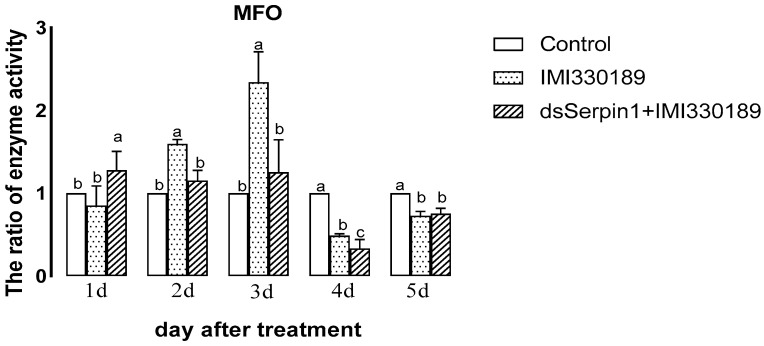
Multifunctional oxidase (MFO) activity of *L. migratoria manilensis* from different treatments. Bars represent mean ± S.E. (n = 3). The letters a, b, and c denote the differences between treatments, bars labeled with different letters are significantly different (one-way ANOVA followed by Duncan’s test, *p* < 0.05).

**Figure 9 insects-12-00178-f009:**
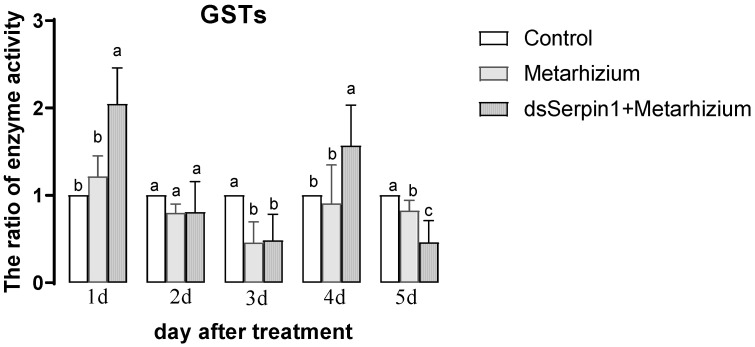
Glutathione S-transferase (GST) activity of *L. migratoria manilensis* from different treatments. Bars represent mean ± S.E. (n = 3). The letters a, b, and c denote the differences between treatments, bars labeled with different letters are significantly different (one-way ANOVA followed by Duncan’s test, *p* < 0.05).

**Figure 10 insects-12-00178-f010:**
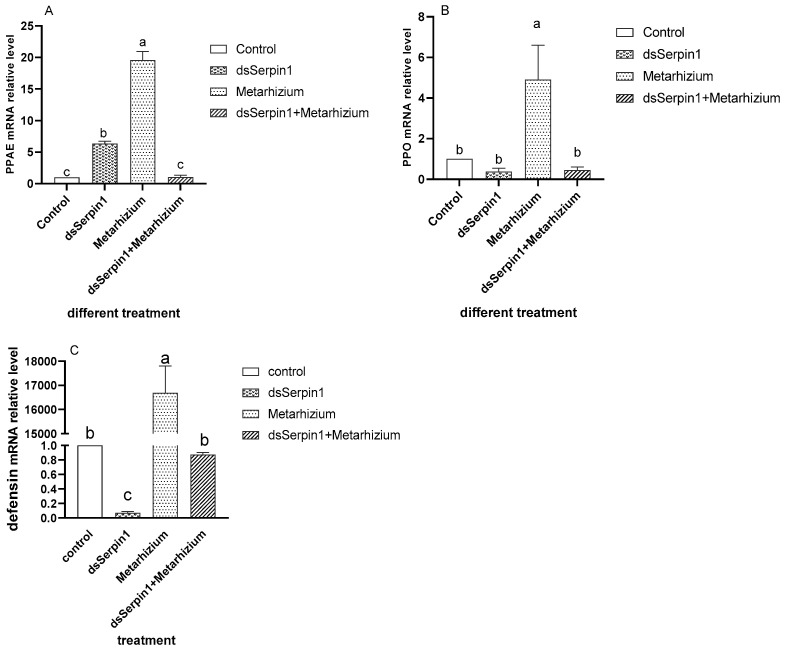
The relative quantitative expression of immune related genes of *L. migratoria manilensis.* (**A**) The expression of PPAE in different treatments. (**B**) The expression of PPO in different treatments. (**C**) The expression of *defensin* in different treatments. The letters a, b, and c denote the differences between treatments, values are expressed as means ± standard error (SE) of the number of treatments. Bars marked by different lowercase letters are significantly different based on least-significant difference (LSD) analysis at *p* < 0.05.

**Table 1 insects-12-00178-t001:** The information of nucleotide sequences of the primers used in this study, and the purpose of corresponding genes. Prophenoloxidase-activating enzyme (PPAE); prophenoloxidase (PPO).

Gene	Primers	Purpose	Sequence (5′–3′)
*serpin1*	serpin1-F	Expression	CGCGGATCC GATGCCAGTCCGCGCCTTCTC
	serpin1-R	Expression	CCC AAGCTT TTGCGGAGGC CTTTGTGG
	qPCR-serpin1-F	Real-Time PCR	TACGCAGGCAAAGGAAAG
	qPCR-serpin1-R	Real-Time PCR	ATGGGTTTACGGTGCTC
	dsSerpin1-F	RNAi	TAATACGACTCACTATAGGATCAGCACAGCCAGGAAAC
	dsSerpin1-R	RNAi	TAATACGACTCACTATAGGCGGCATCGGAGAAGTATTG
*Actin*	actin-F	Real-Time PCR	GTTACAAACTGGGACGACAT
	actin-R	Real-Time PCR	AGAAAGCACAGCCTGAATAG
*PPAE*	PPAE-F	Real-Time PCR	CACCAGCACAAATGAATGAC
	PPAE-R	Real-Time PCR	CAACGACAATGAGGCACAG
*PPO*	PPO-F	Real-Time PCR	AAAGACCGCAGAGGAGAA
	PPO-R	Real-Time PCR	CCAACGATAGAACACAGGA
*defensin*	defensin-F	Real-Time PCR	CCAGAAAGCGATGATGCCACTA
	defensin-R	Real-Time PCR	CACCACAAATCAACGCCAAAGT

**Table 2 insects-12-00178-t002:** The components and concentration of baits from different treatments.

No.	Treatments	The Concentration of *M. anisonliae* (Spore/g Bran)	Dosage of ds Serpin1 RNA (mL)
(1)	Metarhizium	2.50 × 10^8^	0
(2)	dsSerpin1	0	5
(3)	Metarhizium + dsSerpin1	2.50 × 10^8^	5
(4)	Control	0	0

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
