# Peer review of "Molecular Identification and Immunity Functional Characterization of Lmserpin1 in Locusta migratoria manilensis"

_insects, 2021, doi:10.3390/insects12020178_

Round 1

Reviewer 1 Report

In this paper, Li et al. aimed to investigate if Serpin1 was involved in humoral immune responses of Locusta migratoria manilensis, an important insect pest in China, South-East Asia and Pacific region. At this aim, they identified the gene and analysed its expression profiles in different stages and tissues. Furthermore, they use RNAi to interfere with serpin1 gene and evaluate the effects on mortality, the activities of protective and detoxifying enzymes and the expression level of three immune-related genes of L. migratoria manilensis under Metarhizium anisopliae infection.

Understanding the mechanisms allowing pest species to cope with external pressures, such as pathogens is highly important, because some of them can also be used as control tool against them. On the whole, I believe this paper is a valuable contribution to the understanding of the mechanisms underlying the immune responses in insects, but some points should be addressed before publication.

Comment to the Authors:

Serpins is a superfamily including several members. I am interested in knowing why the authors chose Serpin1 gene. The importance of this specific gene has been already highlighted in other species? The authors could add more information about that in the manuscript, to give a more comprehensive picture of what is currently known about this gene and its role in the immune response. Furthermore, they used the fungus Metarhizium anisopliae to infect Locusta migratoria manilensis. The reason behind the choice to use this organism is mentioned only in the discussion, where the authors stated that this species can be used against this pest. I suggest to expand the last paragraph of the Introduction and include more information about these issues.

Lines 109-111: Why did you choose to normalize the expression level on midgut? The expression level likely can change according to what tissue is chosen to normalize. Some information can be added about that, or eventually include some references.

3th instar, 3th nymphs, nymphs with 3th instar: this can be a little confusing. I suggest to choose one of them. Furthermore, some information about the life-cycle can be included in the text.

Figure 2. “Table1 gene” is not clear, maybe it is a refuse? The authors should also specify in the caption that these results refer to RNAi experiments.

Figure 3. correct the caption with “across 12 days after treatment”.

Figure 4. The caption is uncorrected because the PO activity is showed for 5 days after treatment, not only for the 3th day.

Concerning the results about the enzymatic activity (PO, POD, SOD, MFO, GST), I suggest to make a multi-panel figure to portray the data. This would facilitate the reading of the results, also reducing the number of figures in the paper.

Line 26: “Quantitative real-time polymerase chain reaction (qRT-PCR) analysis results revealed that Lmserpin1 25 was expressed in all examined stages and tissues […]”. This is not completely true because as shown in Fig.1B in the hemolymph the expression level of Serpin1 seems zero.

Lines 45-46: this sentence is incomplete.

Line 54: delete eventually at the end of the sentence.

Line 78: Wrote the full name of the species

Line 114: typo

Line 163: typo

Line 164-165: this sentence is not clear, I suggest to rewrite it.

Line 178: change is with it

Line 236, 250, 263, 274, 287: wrote L. migratoria manilensis in italic

Line 237: change increased with increase

Line 325: change defense with defensin

Line 338: move Figure 1B in the line above, after "fat body".

Line 375-380. This sentence is not clear, I suggest to rewrite it

Line 334: the authors stated that the interference efficiency was very high in the hemolymph. According to this, I expected to observe a higher expression level of Serpin1 in the hemolymph of samples that were not exposed to RNAi. In the figure 2, the expression values of the untreated samples were not shown and the only indication about that can be found in the data shown in Fig. 1B, that portrays the constitutive expression level of Serpin1 in different tissues. Here, the Serpin1 mRNA relative level in the hemolymph seems zero. How was the expression level of serpin1 in the samples not exposed to RNAi at 24h? Maybe, the expression level in the different tissues of the untreated samples can be included in the figure 2b.

Line 400-402: This sentence seems contradictory, because Serpin1 seems to increase the immunity of locust and its ability to resist pathogens. Is it true? Maybe, you would intend the interference of the gene?

Line 411: change defense with defensin

Author Response

Please see the attachment “Response to Reviewer 1 Comments”

Reviewer 2 Report

The manuscript by Li et al describes the identification of a previously undescribed serpin in the pest insect Locusta migratoria manilensis. The researchers cloned and sequenced serpin1 and using RNAi, they assessed the role of serpin1 in innate immunity responses to the fungal pathogen Metarhizium anisopliae. They observed that knockdown of serpin1 reduced the ability of the insects to defend themselves against the fungal infection, which supports the prediction that this serpin helps regulate innate immune responses to this pathogen. They observed that several innate immune enzymes, including phenoloxidase, peroxidase, superoxide dismutase, glutathione transferase, and multifunctional oxidase were perturbed by the knockdown of serpin1, and that the expression of three immune response genes were also disturbed by the knockdown of serpin. Not all perturbations were as predicted, and hence, some outstanding questions remain. It is my recommendation that the following points be addressed before publication of this study:

  1. It is unclear how this particular serpin gene was first identified, and how the primers used to amplify the gene were found. The authors should explain how the primers were designed. Was this study preceded by a transcriptomic analysis of this species, or is there a genomic database of this subspecies of locust?
  2. Have other serpins been identified in the locust, and if so, what was the rational for selecting this one?
  3. Is this serpin similar to serpins in other insects? Including a phylogenetic analysis would be useful. Have the orthologues of this serpin in other species been characterized and shown to regulate innate responses to fungal pathogens?
  4. In Figure 1A, there is a large drop in serpin1 transcripts in the 4th instar nymphs, but there is no explanation for why this may have occurred. Have similar drops in serpins been observed during development of other insects? A reduction in the penultimate nymph stage seems rather unusual and some commentary on this should be provided.
  5. In Figure 2, the negative control values, with their variances, should also be included, to clearly demonstrate that the knockdown values are significantly different from the control values. Are the error bars showing standard errors or standard deviations? The number of replicates should also be provided in the figure’s caption, to make it easier for the reader to evaluate the data.
  6. In Figure 2, the level of serpin1 transcripts, following Metarhizium infection, should be shown. Likewise, the extent of knockdown of serpin1, with and without Metarhizium infection should be included in Figure 2. This information will be valuable in the explanations of the results in Figure 9 (see point 12 below).
  7. In Figure 3, the number of insects tested should be included in the figure caption, and the statistical test should be indicated.
  8. What is the significance of “IMI330189”? As someone not familiar with Metarhizium isolates, it would be useful to know if this is a commonly used isolate of this fungus. Instead of labelling the figures with this curious code name, why not simply label the Figures with “M. anisopliae”, or “Metarhizium” and use these words in the body of the text?
  9. The caption in Figure 4 seems inaccurate – shouldn’t it be 3rd instar (not 3rd day after treatment)? Why don’t the control bars have error bars? Is there no variation in PO levels from one individual to the next? On day 4, the control is labelled with a “b”, indicating a significant difference from the other days. Does this mean the error bars were very small? If so, a comment to this point should be added to the figure caption. The caption also needs some further details, including sample size and statistical test.
  10. Figure 5 is not using the labels a, b, and c effectively. As the authors are comparing changes over time, more letters may be needed. For example, day 1 control is labelled with a “c”, but first of all, it is missing an error bar. That value doesn’t look different from days 2-5 controls, but yet it is labelled differently. It does look different than IMI330189 on day 5, which is also labelled with a “c”. Clarification is needed here, or at the very least, errors bars on the controls need to be added. All captions need to explain the meaning of the different letters (i.e. “different letters above the bars denote statistically different values [insert stats test]”).
  11. Line 351. What is Mad1? Some explanation is needed to make this comparison meaningful.
  12. In Figure 9, delivery of serpin1 dsRNA resulted in increased levels of PPAF transcripts but decreased levels of PO transcripts. Can the authors provide a possible reason for this opposing result, as both genes are associated with the same process? Curiously, when the insects are infected with the fungus, PPAF transcripts are downregulated, while they increased with no fungus present. Does this indicate that the fungus prevents the knockdown of serpin1, forcing the insect to override the RNAi effect and actually increase serpin1 transcript levels? See point 6 above to tie this together.
  13. Delivery of dsserpin1 causes reduced PO transcripts (Figure 9) and decreased PO activity by day 2 and 3 (Figure 4). Does the large increase in PO activity on day 4 (Figure 4) indicate that the RNAi effect has worn off, and the insect is now overcompensating for the loss of PO activity? It would be worthwhile measuring serpin1 transcripts on day 4 to address this point.
  14. Can you conclude that serpin1 is a negative regulator of SOD and GST, given that those two enzyme’s activities increase following delivery of dsserpin1? It would be informative to measure these two genes’ transcript levels too.
  15. The Results section would benefit from a more streamlined description of the data, highlighting only the more pronounced changes in enzyme or transcript levels for each experiment, and not getting bogged down in all the minor ups and downs in later days of the infection process (where RNAi may be waning).
  16. The Discussion would benefit from a Table that highlights which responses showed up- or down-regulation of enzyme or transcript levels, coupled with a prediction on whether that implies positive or negative regulation by serpin1, or whether it remains uncertain. Where there is still too much uncertainty about serpin1’s roles, more specific tests should be indicated (stating that “further research is needed” is just too vague).

Author Response

Please see the attachment of“Response to Reviewer 2 Comments”

Reviewer 3 Report

Overall, the authors tested the function of a serine proteinase inhibitor in Locusta migratoria fungal infection. Serpin 1 was interfered and its role in fungal infection was studied.

The experiments were performed with enough replications and the results are interesting and would contribute to understanding the role of serpins in insect immunity.

However, the manuscript need to be significantly improved in writing for clarity. Rationale for choosing only a single serpin protein for this study. There was a potential to study more than one serpin gene with the experimental design. Rewriting the manuscript for delivering the results effectively is strongly recommended before publishing.

Line 20: Abstract is 283 words which is significantly more than the required length of 200 words

Line 22: Expand PPO

Line 23: Replace “stuck” with an appropriate word. This word is anthropomorphic

Line 26: replace signals with signal.

Line 50: Replace were with are

Line 55: Cascades represent a phenomenon where multiple molecules are sequentially involved to activate a pathway like a typical signaling cascade. Here cascades word is not appropriate.

Line 63: Functionally studied?

Line 82: Please give the rationale for selecting Lmserpin1

Line 93: Provide an explanation for metapedes in parantheses

Line 96: replace observed with measured.

Line 111: How was midgut chosen as the calibrator?

Line 126: Replace ddH20 with full form

Line 128: Were starved

Line 136: Replace dynasty with concentration

Line 137: Is wheat bran the diet?

Line 139: It is unclear what 2.3 represents

Line 161: PPAF or PPAE?

Line 203: ANOVA was not mentioned in the statistical analysis

Line 216: Please include the legend that the mRNA levels are after RNAi. Also, why wasn’t the integument included in this experiment?

Line 248: Please include in the legend the explanation of the treatments

Author Response

Please see the attachment of“Response to Reviewer 3 Comments”

Round 2

Reviewer 1 Report

I think that this version of the paper is better than old version, as more results were shown. However, I suggest the authors to pay more attention to the form of the manuscript. There are still many typos and the use of different fonts and sizes along text.

Figure 1. The caption should be corrected. The expression level is normalized on hemolymph, not midgut.

Figure 4. The bars in this figure are not labeled with letters. Maybe, it should be deleted.

Point 4: This point has not been corrected in the text. “Table1 gene” in the caption of the Figure 2 is not clear. Correct it.

Author Response

Point 1: Figure 1. The caption should be corrected. The expression level is normalized on hemolymph, not midgut.

Answer 1:I have changed that the expression level is normalized on hemolymph on figure 1

Point 2: Figure 4. The bars in this figure are not labeled with letters. Maybe, it should be deleted.

Answer 2:I have replaced the line chart with a bar chart and added the differential letters, because it is not easy to add letters on the line chart.

Point 3: Point 4: This point has not been corrected in the text. “Table1 gene” in the caption of the Figure 2 is not clear. Correct it.

Answer 3:Regarding the caption of the Figure 2, we has been changed "Table1" to "Relative mRNA level of the Lmserpin1".

Reviewer 2 Report

The revised manuscript has resolved my concerns with the previous draft. Aside from needing some further editing of the English, I believe that the manuscript will be of interest to those researchers interested in insect immune responses.

Author Response

Thank you. I have checked and revised the English of the revised manuscript